# The Gap between the International Criminal Court and Victims: Criminal Trial Reparations as a Case Study

**Yidou Yang**

Faculty of Law, University of Macau, Macau 999078, China; yc27223@um.edu.mo

**Abstract:** Although victims have the right to limited participation in trials and to seek reparations after sentencing, the legal structure of the International Criminal Court (ICC) prioritizes retributive justice over restorative justice and punishment over reparations. Thus, currently, although the perpetrators can be tried through the ICC, it is still difficult to obtain reasonable compensation for the damages suffered by the victims. On the one hand, the ICC's reparation system may be restricted by the identity of the victim, ICC internal factors, and so on. The current structure of the ICC compensation system allows for hierarchical relationships between victims, while at the same time, there is tension between individual and collective types of compensation. These factors have led to a disconnect and gap between the protection of rights at the theoretical level and actual reparation. This dichotomy between the theoretical protection of the rights of victims and the real protection of victims in practice exists in the ICC. Victims are isolated from the field of vision due to potential repercussions. The idealistic illusion of justice is completed when the ICC stands on the stage and accepts the audience's praise. However, for compensation in criminal courts, people are paying increasing attention to the legal process and content. In practice, the proportion of victims of international crimes is not low, and in some cases, victims are widespread. It can be seen that criminal compensation for victims is an issue that spans a vast range of people and regions. Nonetheless, there are still research gaps regarding reparation and other ideas of justice according to the ICC, how the ICC provides multifaceted safeguards for victims, and the limitations and influence of the mechanism of the ICC on the compensation of victims. Considering the above problems, this paper aims to analyze the International Criminal Court indemnity cases. This paper wishes to analyze reparations and other ideas of justice under the ICC, examining the approach of the ICC toward compensation for victims, where the ICC is heading regarding reparations for victims, how the reparations system works, and the advantages and disadvantages of the reparations system, as well as what are the potential problems of ICC related to reparations. What guarantees do the ICC's mechanisms provide for victims to be able to receive reparations? How does the structure of the ICC reparations system conflict with victims' reparations in practical terms? What are the potential obstacles and gaps between criminal trial reparations and victims? The first chapter wants to analyze the early Nuremberg tribunal, Tokyo tribunal, ICTY, and the ICTR by analyzing whether international criminal justice under these military tribunals was restorative justice or reparation justice and interspersed with analyses of reparation to victims under these tribunals. Then, it analyses it further about justice and reparation of the ICC, and it talks about the compensation for the victim and how the idea of compensation under the ICC has evolved. Using these arguments to analyze reparation and other different ideas of justice under the paly of ICC. The second chapter of the article analyzes the "participatate in trial for compensation", "The limits of participating in trial", "Safety protection for victims" to demonstrate the current protection and progress of the ICC system on the issue of victims' compensation, this is because victims' participation in the trial will bring a lot of help to the issue of compensation. The article analyses the significance and shortcomings of participation in a trial for compensation, which is necessary and meaningful to the issue of compensation because "participation in trial" and compensation are related and complementary to each other, as participation of the victims will bring a lot of help to the issue of compensation. The article analyses the section "Protection of the financial situation of victims: A possible alternative methods of reparation" because, to some extent, it can be seen as an alternative method of ICC compensation. The third chapter of the article hopes that by

analyzing "Little compensation", "The silence court put on victims' rights of compensation", "The ICC's model of judicial administration remains optional" to argue and analyze how the structure of the ICC reparations system conflict with victim reparations in practical terms. Because the silence the court put on victims' rights of compensation and the ICC's model of judicial administration remains optional, both directly impact the issue of compensation. Chapter IV mainly aims to analyze some of the potential negative impacts of the ICC on victim reparations, specifically "The victim's social death", "Restrictions on "expression" between the victim and the court", "Does the ICC hope to improve its attitude to victims?" to specifically analyze and argue these aspects of its potential negative impact on victim reparations. On this basis, this paper analyzes the gap between criminal trial reparations and victims to identify what negatives exist between the two.

**Keywords:** victims; International Criminal Court; criminal trial reparations; gap; reform of mechanism

## 1. Reparation and Other Ideas of Justice According to the ICC

### 1.1. Reparation and Retributive Justice

Restorative justice in international justice inherently requires the participation of victims in the process of trying and repairing the damage caused by the offense. Restorative justice and retributive justice also distribute and influence the mechanisms and development of the ICC's approach to the issue of victim reparations. Therefore, setting out these two approaches to justice, then proceeding with the analysis via a discussion of reparations in international justice, and discussing reparation and other ideas of justice according to the ICC has a direct role to play in a better understanding of the ICC's work on issues related to reparations for victims. The category of international criminal offenses includes crimes such as war crimes, genocide, crimes against humanity, and crimes of aggression, which usually have a wide geographic range of victims and affect an incalculable number of people. The victims suffer irreparable harm to their lives, personal safety, the personal safety of their families, and their property; moreover, they are deprived of their basic rights as human beings, such as property, basic safety, human dignity, and even life. The international community and international criminal justice have called for the perpetrators of these crimes to be punished and for the victims to be compensated, giving rise to the concepts of retributive justice and restorative justice. Retributive justice is a theory that suggests that an offender should be published when he or she breaks the law. It will have a general deterrent effect while punishing criminals. It essentially means repairing justice by punishing the offender. However, it has a weakness: if the focus is only on punishing the perpetrator while ignoring the recovery measures for the victim, the victim will feel that they have been hurt, and there is no corresponding compensation (Tsai 2021, p. 31). Restorative justice, on the other hand, is also an approach to justice for the international community and for the victims which aims to ensure that the perpetrator repairs the harm caused by the crime, which could include apologies, reparations, and other forms of compensation to make amends and reparation for the harm received by the victim (Bonta et al. 1998). It essentially requires the victim and society to be involved in the process of trial and repairing the harm caused by the crime (Canada 2022).

Early international criminal justice as a whole remained, to a large extent, focused on retributive justice. International criminal justice is an area of international law that calls for the prosecution of the most serious crimes, such as war crimes and human rights violations, while at the same time wishing to protect and make reparations to victims (Rodman 2016). The idea of international criminal justice can be traced back to the aftermath of the First World War, and after the Second World War, in 1945, the international community established the first successful international criminal justice institutions, namely the Nuremberg tribunal and the Tokyo Criminal Court (Moffett 2012). In 1945, the Nuremberg military tribunal wanted to provide justice for victims by punishing Nazi leaders, and it was the only compensation that the Nuremberg tribunal offered to the victims. This has manifested

itself in various acts in the courts. In the Nuremberg trials, victims could only participate in the courts by becoming witnesses (Moffett 2012, p. 245) and did not have a voice, let alone the protection of the victim's right to financial compensation, and there is no mention of the victims in the court's establishment documents. The Moscow Declaration (Britannica 2023) only briefly mentions the angry people who have the right to bring a perpetrator to justice, which can be regarded as the only description of the victims. The Declaration of Tokyo (WMA 2023) also made no mention of victims. The only result of the Tokyo tribunal was making Japan admit that it had committed rape and sexual violence against thousands of women, while the Nuremberg trials sentenced 27 defendants to death. Thus, in the early years of the tribunals, it was thought that justice came from judgment of the guilty, and justice meant that millions of angry people had the opportunity to try the perpetrator. With the gradual end of the Cold War in 1991, the international community resumed discussions on the establishment of an international criminal court. Close to this timeline, mass atrocities in the former Yugoslavia and Rwanda prompted the United Nations to establish the International Criminal Tribunal for the former Yugoslavia (ICTY) in 1993 and the International Criminal Tribunal for Rwanda (ICTR) in 1994, two ad hoc tribunals. In 1994, the International Criminal Tribunal for Rwanda (ICTR) was established, two ad hoc tribunals. The International Criminal Tribunal for the former Yugoslavia is governed by the Security Council under Chapter VII of the Charter of the Yugoslavia United Nations ("Action with respect to threats to the peace, breaches of the peace, and acts of aggression"). It was the first case in the history of an independent judicial body dealing with international criminal conduct (Deschênes 1994, p. 269). The ICTY and ICTR began to include a definition of victim. With regard to the question of compensation, the Trial Chamber could discuss related issues, such as restitution of property only if the perpetrator had been convicted and the sentence contained evidence that the perpetrator had stolen the victim's property. There was no issue of additional compensation at that time. Compensation is limited to the return of stolen property in the ICTY and ICTR. Although the ICTY and ICTR have made progress on the issue of victim compensation, it can be seen that early international criminal justice was still focused on retributive justice. Subsequently, the establishment of the ICC under the Rome Statute was the crystallization of the international community's long-term desire to hold the perpetrators of serious crimes accountable the perpetrators of serious crimes (International Committee of the Red Cross 2021). In general, international criminal justice did not cover the issue of victim compensation until the statute of the International Criminal Court was formulated (Dwertmann and Ebrary 2010, p. 25).

The ICC's initial preference was to focus on retributive justice and gradually turned to various restorative measures for victims. In 1998, the idea of a permanent International Criminal Court was subject to a series of negotiations, and in July 1998, the Statute of the International Criminal Court was adopted. The ICC was established under the Rome Statute. Although the whole of the early ICC was about retributive justice, victims could not obtain compensation. The ICC's initial preference was to focus on retributive justice. Overall, it was a retributive dispute, and there was no concept of reparation for victims in the early years of the ICC. Due to the disadvantages of vindictive justice, the ICC originally focused too much on the characteristics of vindictive justice, which greatly inconvenienced victims. Therefore, in the subsequent development of the ICC, it gradually turned to various restorative measures for victims, including economic compensation and other measures, which became a highlight and feature of the development of the ICC, and then developed a series of classic international criminal justice organizations such as the TFV. As a result, the development of the ICC has become highly anticipated and of general international interest.

### 1.2. Approach of ICC toward Compensation for Victims

As reparations are increasingly seen as the standard for restoring harmony between warring parties and warring communities, the courts have gradually begun to consider reparations for victims, and its consideration of reparations has begun to move slowly in

the direction of thoughtfulness (Keller 2006, p. 191). For example, the Ntaganda[1] Trial Chamber stipulated that repairing the harm inflicted on victims should be the first consideration when determining the amount of compensation required by perpetrators, which is typical of restorative justice. Moreover, the court has a special trial for compensation at the trial stage. The victim can participate in the compensation trial, become a party to the court, and present evidence against the perpetrator. After the ICC makes an order for compensation, the offender's property will be enforced using the courts of other countries (Sacerdoti and Acconci 2011). Moreover, judging from the current court rulings, the court is considerate in its compensation for victims. In the Katanga case[2], the Trial Chamber awarded collective awards to all victims, in addition to the symbolic award of USD 250 awarded to each victim. Collective compensation includes housing assistance, education assistance, income-generating activities, and psychological rehabilitation, mainly aimed at repairing the damage inflicted on victims of crimes and helping them return to normal life. Although, in theory, courts do not exclude individual restitution and offer individual restitution to victims, such restitution is rare. In the Katanga case[3], the court gave symbolic compensation, which is ostensibly personal compensation, but it is still similar to collective compensation in essence.

Based on the compensation orders issued by the ICC, the compensation method is relatively fixed, which is the combination of symbolic compensation and collective compensation. To date, the ICC has issued only four victim compensation decisions: the Al Mahdi case[4], the Katanga case[5], the Lubanga case[6], and the Ntaganda case[7]. Since there are few compensation judgments, there is very little to study in its compensation development. In the Katanga case, the Appeals Chamber of the ICC noted that the contribution of other individuals to damages is irrelevant in terms of the amount of liability of the convicted person. The Appeals Chamber wishes to remove obstacles to the enforcement of compensation proceedings. In this case, the Trial Chamber awarded symbolic compensation of USD 250 each to 297 victims and made four collective restitutions to all the victims. The order may not have been welcomed by victims where compensation awards have long been difficult to enforce, so it would have been quite a shock to people when the Trust Fund's Board decided to provide USD 1 million in compensation to the victims.[8] It can be seen that the Trust Fund paid the compensation judgment on behalf of the offender. The practice of the TFV in compensating for the offender is quite novel, which is a remedy for the court's difficulty in executing the offender's property for compensation. In this way, it can significantly reduce the victim's negative evaluation of the court and improve the victim's support for the court. This measure may provide a solution for the court to resolve other conflicts between itself and the victim. Subsequently, in the Lubanga case of 2017[9], given that Mr. Lubanga's poverty might not have been able to compensate the victims promptly, the Chamber asked whether the TFV could designate an additional sum for the implementation of collective compensation. This is similar to the Katanga case, which ultimately asked the TFV to raise funds to repay the victims. The Al Mahdi case in 2017 was also similar to the previous two cases. This means that there is a paradigm

---

1　The Case of The Prosecutor v. Bosco Ntaganda' (Moffett 2012), Trial Chamber II, 30 March 2021, ICC-01/04-02/06 (International Criminal Court 2008b).

2　The Case of the Prosecutor v. Germain Katanga',Trial Chamber II, 7 March 2014, ICC-01/04-01/07 (International Criminal Court 2022).

3　See footnote 2 above.

4　The case of the Prosecutor v Ahmad Al Faqi Al Mahdi', Trial Chamber VIII, June 2012, ICC-01/12-01/15 (International Criminal Court 2015a).

5　See footnote 2 above.

6　The case of the Prosecutor v Thomas Lubanga Dyilo, Trial Chamber II, 1 December 2014, ICC-01/04-01/06 (International Criminal Court 2023b).

7　See footnote 1 above.

8　The case of the Prosecutor v. Germain Katanga, Redacted version of "Order For Reparations Pursuant To Article 75 Of The Statute", Trial Chamber II, 24 March 2017, 1 December 2014, ICC-01/04-01/07-3728-tENG (Trial Chamber II 2017).

9　See footnote 6 above.

of compensation: in the four existing compensation cases, the TFV participated in the repayment of the compensation order, and the compensation mode is relatively fixed, which is a combination of symbolic compensation and collective compensation. However, the court did improve on the previous cases. Unlike in the Katanga case[10], the Chamber wanted the TFV to contact the DRC government to explore how it could have contributed to the compensation process (ICD 2012).

Furthermore, the ICC is not accepted by some countries, making it difficult to enforce compensation orders in some cases. In the cases Prosecutor v Ali Muhammad Ali Abd-Al-Rahman[11] and Ahmad Muhammad Harun[12] and Prosecutor v Omar Hassan Ahmad Al Bashir[13], the suspects, Kushayb, Harun, and Al Bashir, are Sudanese nationals. The court issued five arrest warrants in these cases, but Sudan refused to comply with the ICC's orders. Sudan gave the reason that the ICC would violate its sovereignty and that the Sudanese people should not be tried by the ICC (Bekou and Birkett 2016, p. 24). Such reasons and phenomena are not surprising. In practice, most countries do not refuse to cooperate outright but rather do not act or delay (Bekou and Birkett 2016, p. 67). Another example, in December 2014, the ICC threw out the case Prosecutor v Uhuru Muigai Kenyatta. Fatou Bensouda, the prosecutor in the case, agreed that the case could not proceed. The Kenyan government failed to cooperate with the prosecutor's office in obtaining financial records and other evidentiary materials from President Kenyatta (Bekou and Birkett 2016, p. 115). Moreover, although State parties to the ICC are obliged to cooperate with the ICC in the investigation and execution of crimes, once the offender has fled to other countries, which are not within the scope of the State parties, it will take a lot of time and effort for the ICC to communicate with the country about the arrest, seizure, and freezing of funds and a result is not guaranteed (Bekou and Birkett 2016). Therefore, it takes a lot of time and energy for the ICC to execute the compensation judgment, which is very difficult and not guaranteed; it can also be argued that the success of the court's execution depends on whether the ICC is fortunate enough to receive help from the countries concerned. However, it is wrong to be too pessimistic about the ICC because the court has the right to find non-compliance in such cases. In short, other States can resist the decision to cooperate with the court, and the court, as an independent international organization, is entitled to respond to the corresponding non-cooperation; courts still struggle to enforce the evolution of compensation orders.

### 1.3. How the Idea of Compensation Has Evolved

Expanding the scope of compensation for victims may be the trend of the development of compensation orders. In the Ntaganda case (Kuniewicz 2015), Trial Chamber VI of the ICC sentenced Mr. Ntaganda to be liable for compensation for a total amount of USD 30,000,000. The courts award compensation to indirect victims, depending on the nature of the injury. Compensation is no longer limited to immediate victims and their families. The Chamber noted that, for victims of reparation, priority should be given to individuals who are in need of immediate medical care, are disabled, have suffered sexual or gender-based violence, or are homeless. This has not happened in the previous three cases. Thus, it may be a trend toward compensation enforcement. Although the case is still in trial and the compensatory judgment is not final, the development of the court's view on compensation can be studied from this case. According to the tendency of the court to gradually expand the scope of victim qualification, the expansion of compensation scope may be the trend of the court's development.

---

10　See footnote 8 above.
11　The Prosecutor v. Ali Muhammad Ali Abd-Al-Rahman, Trial Chamber I, ICC-02/05-01/20 (International Criminal Court 2023a).
12　The Prosecutor v. Ahmad Muhammad Harun, Pre-Trial Chamber II, ICC-02/05-01/07.
13　The Prosecutor v. Omar Hassan Ahmad Al Bashir, Pre-Trial Chamber II, ICC-02/05-01/09.

### 1.4. Two Approaches to Compensation: Collective and Individual Compensation Methods

The advantage of collective reparation is that it is flexible, which includes being more adaptable to large-scale crimes. Collective reparations are collective measures taken by victims to achieve compensation.[14] In the Al Mahdi case, the court demanded an apology in relation to the buildings. However, these offers were rejected by the victims as a form of compensation. Similarly, victims in Katanga also rejected collective symbolic measures. In contrast to collective symbolic compensation, the reason for the victims to try to obtain compensation can be determined to be a need for substantial compensation. The number of victims of persistent violent crimes is large, often on a community or regional basis. Faced with this situation, it is difficult for the court to award individual compensation, and such a process would be complex and protracted (Moffett and Sandoval 2021, p. 761). The use of collective reparations can, therefore, avoid cumbersome procedures and burdensome tasks for the ICC, and individuals who are not eligible for victim compensation are more likely to offer collective compensation. Some victims find it difficult to access the individual compensation system, so collective compensation is better for them (Lambert 2017, p. 9). Similarly, collective compensation usually includes an individual compensation component in practice but is not referred to as individual compensation. In the Ntaganda case[15], the court states that collective reparations should include individualized components. The court grants pensions or living expenses to individuals who are unable to participate in collective compensation, which is a form of compensation, even if it is not called personal compensation by the court. This is essentially an application of the flexibility of the court to collective compensation. Flexibility is also reflected in the different needs of different victims, and the court has different ways to compensate them. In the Katanga case, the Chamber and the TFV adopted flexible compensation methods. Childless victims were no longer compensated for educational support but received other benefits (Moffett and Sandoval 2021). Fortunately, collective reparations can help repair the communities in which the victims live. This could mean restoring infrastructure to the communities where the victims live and providing education services to the children of the victims. This approach could significantly increase the satisfaction of the victimized communities with the ICC. However, in the case of individual compensation, monetary compensation is usually symbolic and can cause the victim to complain to the court. In this case, relatively speaking, collective compensation is more in line with the interests of the court. As in Katanga with the Trial Chamber, collective compensation was provided to the victims, including housing assistance, income assistance, psychological rehabilitation, and other benefits.

Collective compensation has the same advantages that individual compensation has but also has a convenience that individual compensation does not have. The disadvantage is that the court's convenience for collective compensation, as well as its consideration of the cost of compensation, can vary widely from region to region. For example, groups close to the area where the overall crime occurred were more likely to receive higher compensation. The reason is the proximity of the area to where the overall crime occurred, and it is convenient for the court and the TFV to provide compensation services. The distance from the area where the overall crime took place makes it difficult to find refugees fleeing to Europe or the United States, and the distance makes it difficult to enforce compensation. Specifically, in the Katanga case, 15 of the 297 victims eligible for compensation in the Democratic Republic of Congo were living in Europe or the United States. These 15 people were separated from the collective, so it was difficult to compensate them collectively. The TFV found it impossible to distribute any compensation to the victims who live in the United States and Europe; the courts and the TFV sometimes provide less or no compensation to victims who are far away for the convenience of execution (International Criminal Court 2017).

Moreover, the collective reparations modalities do not include monetary compensation. Although there are exceptions where some beneficiaries cannot be compensated by

---

14 See footnote 8 above.
15 See footnote 1 above.

means of collective compensation, where the court may make an exception for monetary compensation, these are very exceptional cases. For most victims of collective compensation, monetary compensation is not available. It is clear that in some cases, victims whose families have been killed or who have lost their means of support are in greater need of financial help (Lambert 2017, p. 13). In the Katanga case, the victims were not satisfied with other types of compensation and recommended that the Trial Chamber use part of the perpetrator's income and assets for victim compensation. The victim has clearly shown a preference for financial compensation in this case.

*1.5. Conclusions*

Early international criminal justice as a whole remained, to a large extent, focused on retributive justice. Subsequently, the establishment of the ICC under the Rome Statute was the crystallization of the international community's long-term desire to hold the perpetrators of serious crimes accountable the perpetrators of serious crimes. The ICC's initial preference was to focus on retributive justice, and in the subsequent development of the ICC, it gradually turned to various restorative measures for victims, including economic compensation and other measures, which garnered a lot of attention and raised the expectations of the international community which have formed its highlights and characteristics. As reparations are increasingly seen as the standard for restoring harmony between warring parties and warring communities, the courts have gradually begun to consider reparations for victims, and its consideration of reparations has begun to move slowly in the direction of thoughtfulness. Based on the existing four compensation judgments, the compensation method is relatively fixed: collective compensation combined with symbolic compensation. The advantage of collective reparation is that it has greater flexibility, including adaptability to large-scale crimes. More importantly, collective compensation can significantly rehabilitate the victim's, or victims', community and help the group return to normal life, which is of great significance to victims. However, individual compensation can ensure that the victims secure more monetary compensation, which is exactly what collective compensation cannot achieve, and the victims are more inclined to receive economic compensation.

## 2. How the International Criminal Court System Can Provide Guarantees for Victims' Compensation

*2.1. Participating in Trial for Compensation*

Over time, the ICC allows victims to become a party in reparation trials and to be witnesses, which is very meaningful for compensation because victims' participation and compensation are related and complementary to each other, as participation of the victims will bring a lot of help to the issue of compensation (Miller 2012). The term "reparation"[16] in the ICC Statute refers to the court ordering the convicted criminal to pay compensation to the victim (Dwertmann and Ebrary 2010, p. 10). In the PROSECUTOR v MAHAMAT SAID ABDEL KANI case, the Trial Chamber VI issued a decision authorizing 20 victims to participate in the trial.[17] Hence, compared with the ICTY and ICTR, the ICC allows victims to participate in the trial itself, which is an affirmation of victims' compensation rights since participating in the courts allows victims to press for more damages and more power to fight the perpetrators.

The court of the qualification of the victim is not strict, only a superficial review, which is very favorable to victims' compensation matters. Even if victims can assist prosecutors, the impact on prosecutors is limited. In addition to making complaints and opinions, providing relevant evidence and complaints basically cannot play an important role (Sanders 2002, pp. 4–5). As a result, the courts have offered victims other remedies. For example, in the Ntaganda Trial Chamber, when the victim provides evidence to the

---

16　Art. 75 (2) of the ICC Statute.
17　The case of the Prosecutor v. Mahamat Said Abdel Kani, Redacted version of "Decision authorising 20 victims to participate in the proceedings", 27 May 2022, Trial Chamber VI, ICC-01/14-01/21-331 (International Criminal Court n.d.).

court, the court is not limited to the form of evidence; the victim can provide documents, and no document can be signed by two credible witnesses. Moreover, the qualification of victims only needs superficial examination. In the Prosecutor v Mahamat Said Abdel Kani case[18], the prosecutor, the defense, and the Office of Public Counsel for the Victims (OPCV) argued bitterly over victims who were eligible to attend the trial. The OPCV believes that the perpetrator's crime left the victim with traumatic scars, which affected the accuracy of the victim's memory of the event. The court should adopt a flexible approach to ensure the interests of victims. The prosecution argued that a victim without evidence was not entitled to victim status. The Chamber stressed that the criteria for identifying victims should be a preliminary assessment. It rejected the defense's approach, arguing that there was no need for a systematic and in-depth assessment of the information. The Court further noted that the victim application procedure was not a mini-trial procedure and that whether the victim had genuinely suffered harm would be judged at the time of conviction.[19] It means that the court's identification of the victim's qualifications is not strict but a superficial examination. The court's surprise decision shows a lax attitude toward victims that is unlikely to change in the future. More victims will be recognized as eligible to receive compensation, and once qualified to participate in a trial, victims will be able to directly participate in the trial process, such as appeals, without having to apply, which means victims can participate in the whole trial (Perez-Leon-Acevedo 2018). Indeed, there are some drawbacks to this tendency. The lax attitude of victim qualification may lead to the participation of some false victims in court, and thus, there is a risk of inauthentic evidence being admissible, which, in a sense, violates the rights of perpetrators.

Evidence of the crime only needs a reasonable acquaintance with the facts of the case. As long as the credible statement of the victim and the presumption of fact are reliable, the fact of the injury can be presumed. For instance, rape victims do not need to be scrutinized for specific injuries and may be considered eligible as victims as long as their statements and presumptions relating to the facts are reliable. The deceased may be represented in court by his or her immediate family.[20] Even if a natural person dies in the course of proceedings, the international court of criminal law allows compensation proceedings to continue as long as the immediate family of the deceased is represented.[21] Moreover, concerning the proof of victim participation in the court, the court takes into account the fact that it is often difficult for victims to produce complete and sufficient evidence to prove consistency. The ICC has adopted a flexible review. In the Bemba case, for example, the lack of a date for the event did not have a serious impact. The court usually makes a judgment based on the overall information presented.[22] It is victim-friendly. Misrepresentation of relevant facts in the application does not affect consistency. Although some scholars believe that many victims are in relatively underprivileged areas and do not have access to educational opportunities (Manirabona and Wemmers 2013), many victims struggle to fully articulate their needs and the degree of damage when filing claims, which affects the ability of victims to obtain compensation. However, due to the assistance of the Victims Participation and Reparations Section (VPRS), OPCV, and other organizations, most victims can fully articulate the details of their injuries via relevant organizations of the court, and therefore, low education is not a large influencing factor.

---

18  The Prosecutor v Mahamat Said Abdel Kani [2019] International Criminal Court, ICC-01/14-01/21 (International Criminal Court).

19  The Prosecutor v. Mahamat Said Abdel Kani, version of "Public Redacted Version of "Prosecution's first request to introduce prior recorded testimony pursuant to rule 68(3)", 31 May 2022, ICC-01/14-01/21-322-Red; The Prosecutor v. Mahamat Said Abdel Kani, 27 May 2022, Office of the Prosecutor, ICC-01/14-01/21-326-Red (Office of the Prosecutor 2022).

20  Prosecutor v Laurent Gbagbo; The Case of The Prosecutor v. Bosco Ntaganda' (International Criminal Court 2022), Trial Chamber II, 30 March 2021, ICC-01/04-02/06.

21  Prosecutor v Jean-Pierre Bemba Gombo, ICC-01/05-01/08; Prosecutor v Ngudjolo Chui, ICC-01/04-02/12 (International Criminal Court 2008a).

22  Prosecutor v Jean-Pierre Bemba Gombo, ICC-01/05-01/08; Prosecutor v Dominic Ongwen, ICC-02/04-01/15 (International Criminal Court 2005).

In addition to broadening the scope of victim qualification criteria, in the victim application process, applications are assisted by the public information outreach section (PIOS) and the VPRS. The PIOS itself contacts affected groups via its outreach activities. Using communication and contact with victim groups, courts are initially informed of the situation and needs of victims and take appropriate measures to assist organizations. In addition, the outreach agency acts as an advocate for the ICC among local people. In 2007, for example, outreach agencies helped increase awareness of the ICC in Uganda from 25% to 60%. Thus, outreach groups can help local people understand the ICC's mission and how it works (Roach 2013). Such participatory justice would be incremental progress.

### 2.2. Limits of Participating in Trials for Compensation

The shortcomings of the VPRS significantly impede victim participation in court. The VPRS, due to logistical and security challenges, has always relied partly on intermediaries to fulfill some of their responsibilities.[23] The court also does not approve of the practice. Pre-trial Chamber II believes that the VPRS should not rely on intermediaries to complete the victim's application but should directly contact and assist the victim in completing the application form. Due to the underdevelopment of some areas and the breadth of affected communities, the two major criteria of the VPRS are generally accepted via the assistance of retailers.[24] However, this approach is not very reliable. For some of the victims, it is difficult to contact the remote VPRS, and it is difficult to obtain help from mediators. Moreover, it is often difficult for a victim to tell a detailed story with an intermediary or to share the extent of their victimization with an influential leader in the community. Second, it is hard to say whether there is a problem with this indirect communication. As the VPRS is an organization dominated by Western culture, it is difficult to truly understand the significance and importance of the cultural customs in the area where the victims are located. Under current ICC practice, language experts usually translate the victim's evidence and material and deduce what the victim wants to say. Since VPRS evidence may be gathered using outreach agencies, language experts do not have the opportunity to communicate directly with victims, so miscommunication remains an issue. The VPRS also simply does not have enough money to conduct extensive checks on the evidence it collects.

Furthermore, due to victims usually appearing in groups, the court selectively involves a very small number of victims as individuals. As an illustration, the Trial Chamber in Bemba selected three victims from the victim group to participate directly in the court. It is highly selective. This high selectivity goes against the will of the victim and objectifies him or her. Scholars have argued that the victims are the ICC's "Brand". The ICC hopes to shape an institutionalized concept of the ideal victim for all member states, making this institutionalized concept a "spokesperson" for global justice and a brand for the ICC to promote and enhance its influence. In the ICC visitors center, for example, the ICC presents its work in a series of images, text, and sound. In the "Victim Exhibit" section, there is a recording of the "victim" talking about how testifying made him or her feel free and proud of a job well done. However, this is not from the actual victim; it is the ICC using its constructed victim identity to promote itself (Schwöbel-Patel 2021, p. 147). If the individual victim wants to increase the protection of his or her rights, he or she needs to compete with other victims. In addition, in practice, the voices of those victims who have not been allowed to participate directly in the trial must be mediated using lawyers (REDRESS 2005), and victims' lawyers are not always given the right to speak in court (Haslam 2011) because of restrictive rights of audience (McEvoy and McConnachie 2013, p. 495). Additionally, while the opinions of these victims' lawyers may have given the court a better understanding of what happened (Garbett 2013, p. 207), it is difficult to discern how their opinions actually

---

23   See footnote 1 above.
24   See footnote 1 above.  See also the case of the Prosecutor v Dominic Ongwen, ICC-02/04-01/15 (International Criminal Court 2005).

affected the final judgment (McEvoy and McConnachie 2013, p. 495). The ICC should be the core of the international community's condemnation of international crimes. It should not be developed to favor the court's preferences. This will damage the stability of the international community and the responsibilities entrusted to the ICC by that community. The question of the court system may not represent the slanting of the court's objectives, but at least measures should be developed as far as possible to avoid such deficiencies (Schwöbel-Patel 2021; Sander 2019).

### 2.3. Safety Protection for Participating in Trials for Compensation

Victims often refuse to participate in the courts for fear of being threatened by the perpetrators, who often have power and even run amity in a country. Japan's Yasukuni Shrine still includes the names of 14 class-A war criminals, which shows that these war criminals were honored by Japanese political circles even before they were executed, and many of them were political leaders. Most perpetrators are capable of committing international crimes with impunity, inevitably backed by national forces. To address the victims' reluctance to give evidence against the perpetrators, courts set up protection mechanisms. The ICTY has established a victims and witnesses unit for victims to support and protect victims. The ICTY's original approach was necessary. In ICTY trials, for example, it is common for witnesses to receive intimidation and anonymous phone calls, while in the case of PROSECUTOR v ALI MUHAMMAD ALI ABD-AL-RAHMAN[25], the court expressed concern that the witness P-1047 may have been at risk from members of Sudan's past and current government who feared being associated with the Sudan Darfur conflict.[26] Thus, the court put the witness under court protection.

Hence, the court will keep the identity of witnesses confidential depending on the circumstances. Because witnesses may be threatened by defendants as well as their lawyers, the court may relocate witnesses to other countries to avoid such threats (von Wistinghausen 2013). for example, one percent of witnesses testifying before the ICTY were relocated to a third country.[27] Coupled with this, the court will be anonymous. Victims do not have to be physically present at the courthouse, and their legal representatives will listen to their views and concerns throughout the proceedings. The identities of the victims will be protected by the courts, and they will remain anonymous or use pseudonyms at all times. (The court even hides all information or content that might expose the witness from relevant court documents and treats documents related to the witness as confidential information[28]).

For example, in Uganda, the TFV participates in the investigation of victim information. Most of the victims to whom the ICC and civil society have spoken say they want to keep their information private. However, with the development of the ICC, an increasingly prominent problem is that the Court will participate in the compensation of the affected areas by using the outreach agencies and TFV, which will visit the victims' areas to investigate the compensation and other issues. As a result, the victim's privacy may not be fully protected. Their identities are hidden from the public because they fear being threatened or killed by perpetrators who have not yet been arrested. However, at present, the outreach agencies and TFV do not have relevant provisions to protect the privacy of victims, and the relevant provisions only exist in the process of court trial, which leads to insecurity for many victims (Roach 2013).

### 2.4. Protection of the Financial Situation of Victims: Possible Alternative Methods of Reparation

The court focuses on protecting victims by providing them reparative assistance and financial support, which, to some extent, can be seen as an alternative method of ICC compensation. This is mostly restorative justice. Financial support for victims exists mainly

---

[25] The Prosecutor v. Ali Muhammad Ali Abd-Al-Rahman, Trial Chamber I, ICC-02/05-01/20.
[26] See footnote 11 above.
[27] Information obtained from VWS.
[28] See notes footnote 22 above.

because most of the victims are poor. They cannot partly afford normal support and find it difficult to hire lawyers to participate in court. Thus, within the Court, the Office of Public Counsel for the Victims (OPCV) will provide legal services for the victims throughout the whole process, which avoids the obstacle of most economic problems faced by victims. For example, in the Prosecutor v Mahamat Said Abdel Kani case[29], the OPCV submitted relevant documents on the victims' observations[30] on the review of Mr. Said's detention to the tribunal.[31] By contrast, the OPCV cannot represent the views of every victim. The positions of OPCV-appointed legal representatives are sometimes at odds with the wishes of victims and may not even represent the positions and opinions of the majority of victims (Sander 2019, n6). This is because the OPCV usually has to deal with a large number of victims and has a great deal of power, which leads to a certain degree of arbitrariness (Killean and Moffett 2017; Haslam and Edmunds 2013). However, the OPCV can mainly represent the interests of victims as a whole, who would otherwise be able to bring the issue to court. In addition, it is easier to use restorative compensation provided by the court to help victims of social marginalization return to their families and society. The ICC can provide education and psychological counseling (Lambert 2017, (n30)10). For example, during the Second World War, the German army captured a large number of women as "comfort women". These women inevitably suffered from physical pain and disease. Most cruelly, they were abandoned by their societies after being sexually abused. They were insulted and mocked by most people and lived on the margins of society. If they could be helped to settle in a new place, have a career, and gain the courage to face life again, it would give them a second chance at life. This is actually not uncommon in international criminal offenses. In the Rwanda genocide, victims were physically and mentally destroyed, the disappearance of their homes and the conditions in which they were able to survive. In the Katanga case, the court awarded four collective restitutions, focusing on psychological support, educational assistance, income generation assistance, and housing assistance. It is significant and necessary for the ICC to provide assistance to help victims reintegrate into society.

*2.5. Benefits of TFV for Victims in Compensation Prospect*

Collective compensation will involve the cooperation of multiple governments, which will take a lot of time, and it is uncertain whether negotiations between governments and courts will be successful. Individuals are unable to initiate collective compensation on their initiative, nor are they able to control the process and procedures. The victim's only option is to defer to the courts, even if that means waiting indefinitely, whereas TFV can greatly increase the scope for victims to receive compensation (Trumbull 2007). TFV, which is funded independently of the courts, was originally set up to create a fund for victims and their families (Ingadottir 2003, (n69)111), and it is recognized internationally as the most effective organization to promote the ICC to realize reparations. Even though the ICC purports to offer compensation only to victims of alleged crimes, this is not the case. The TFV's funds will be used directly to compensate victims rather than waiting for a verdict. Moreover, the TFV is financed not only by compensation judgments against perpetrators but also by social contributions to the fund, as well as reparations and fines transferred by Article 79 of the Rome Statute (UN General Assembly 1998) and Article 98 of Rules of Procedure and Evidence[32], which can cover compensation for victims and their families (Ingadottir 2003, p. 111). For this reason, the TFV can go beyond the trial and judgment of the court by directly using a wide range of funds to compensate victims and their families, which effectively expands the scope of victim compensation and enables victims to achieve the fastest remedy.

---

[29]   Prosecutor v Mahamat Said Abdel Kani.

[30]   Victims' observations on the review of Mr. Saïd's Detention. Available online: https://www.icc-cpi.int/court-record/icc-01/14-01/21-336 (accessed on 13 July 2023) (Office of Public Counsel for Victims 2022).

[31]   The Prosecutor v. Mahamat Said Abdel Kani, Appeals Chamber, 19 May 2022, ICC-01/14-01/21-318.

[32]   Rules of Procedure and Evidence, Published by the International Criminal Court, ISBN No. 92-9227-278-0.

However, there is an irreconcilable contradiction between the ICC and the TFV in the execution of compensation orders. To date, the ICC's compensation order has not been implemented (Ingadottir 2003, p. 14). In fact, the statute of the ICC refers only to the ICC in terms of financing, so the TFV and the court's budget system are separate and mutually exclusive. In individual awards, when the court orders the offender's property to be used to compensate the victim, even if the offender's property is provided with the TFV[33], the money will be transferred to the court instead of being directly allocated by the TFV to the victim (Dwertmann and Ebrary 2010, (n7)267). It can be seen that the TFV and the court are two independent individuals in terms of funds. Therefore, neither voluntary contributions stipulated in Article 116 nor budget contributions of other ICC organizations could be allocated to TFV (Dwertmann and Ebrary 2010, (n7)266). Of course, in the case of individual awards, TFV as a whole needs to be assigned and enforced by the court. However, this situation is reversed in the case of collective awards. In the case of collective award, the TFV has the right of partial decision and supervision over the execution of the court. At present, most of the compensation judgments of the court are made in the way of collective compensation, so in practice, the TFV and the court have different opinions on the implementation. In collective awards, the contradiction between the two institutions arises from the different concepts of compensation distribution. The TFV not only distributes compensation according to the court's compensation order but also considers factors such as the nature of the crime and the degree of injury to the victim. Different perceptions of enforcement between the two agencies have led to delays in compensation for victims. This circumstance was found in the Lubanga case[34]; the court approved the compensation in 2016, but the TFV suspended the assessment of the Harm Assessments. It considered that the Trial Chamber's decision was harmful to the victims. Thus, when the TFV conflicts with court enforcement, enforcement of restitution is extended indefinitely, and only the victim is threatened with indefinite delay of the order. It is encouraging that the TFV and the ICC share the same goal to some extent; that is, they both want to meet the medical, psychological, or basic life needs of the victims of crimes as soon as possible. However, the question is whether the courts are willing to do so, and the choice is entirely in the hands of the courts. Thus, most of the actions of the TFV need the approval of the court. If the two opinions are not consistent, the execution of the judgment will be deadlocked. Even if the courts share some of the same goals as the TFV, it still depends on what the courts ultimately decide.

*2.6. Conclusions*

The court provides many safeguards for victims. First, it allows the victim to participate in court and even become a party to the compensation trial. Although victims' access to the International Criminal Court remains limited, it offers them a variety of remedies. For example, the detachment of the qualification of victims to carry out a superficial examination substantially expanded the scope of the qualification of victims. Nonetheless, courts should consider the risk that this may also violate the human rights of the perpetrators. In addition, the activities of PIOS, VPRS, and other institutions also assist victims. The VRPS helps victims apply, and participatory justice of this kind would be an incremental step forward. However, it is important to note that the VPRS has always relied on outreach organizations to contact victims, which means there is also a layer of intermediary organizations between the court and victims. Since the VPRS is a Western-dominated organization, it is difficult to truly understand the meaning and importance of cultural practices in the area where the victims lived. Thus, it is hard to say whether there is a problem with this indirect communication. Moreover, the court also needs to translate the victim's evidence collected by the VPRS again, which makes it difficult for the court to com-

---

33 Art. 75 (2) 2. Alt., Rule 98 (2) and (3).

34 Situation in the Democratic Republic of the Congo, The Prosecutor v Thomas Lubanga Dyilo, Trial Chamber II, ICC-01/04-01/06.

municate with the victim. Most of the communication is based on the inference of experts. In addition, because victims usually appear as a group, courts selectively present a very small number of victims as individuals. Second, the court provides extensive protection for the safety of victims, which is very important because victims' participation in the trial will bring a lot of help to the issue of compensation, and the court keeps the identity of witnesses confidential on a case-by-case basis. However, with the development of the ICC, an increasingly prominent problem is its involvement in the compensation of the affected areas using outreach agencies and the TFV, which visits the victim's area to investigate the compensation and other issues. As a result, victims' privacy may not be adequately protected. Additionally, the ICC protects the financial situation of victims, which is also a possible alternative method of reparation. The court focuses on protecting victims by providing them reparative assistance and financial support, which, to some extent, can be seen as an alternative method of ICC compensation. For example, OPCV will provide legal services for the victims throughout the whole process. The disadvantage of OPCV, however, is that it does not represent the views of every victim, and perhaps not even the positions and opinions of the majority of victims, and that this leads to a certain degree of arbitrariness. In addition, the restorative reparations provided by the courts are important and can be seen as a possible alternative method of reparation. For instance, psychological support, educational assistance, income generation assistance, and housing assistance. Furthermore, the most anticipated is the appearance of the TFV. Since individuals can neither initiate collective compensation nor control the process and procedure of compensation, the only option for victims is to obey the court, even if that means waiting indefinitely. Currently, the TFV can greatly increase the scope for victims to receive compensation. The TFV can go beyond the trial and judgment of the court by directly using a wide range of funds to compensate victims and their families and effectively expand the scope of victim compensation so that victims achieve the fastest relief. However, the TFV's compensation is conditional on its conduct being approved by the court, which means the decision is back in the hands of the court. To date, it seems that the issue of victim compensation will eventually return to the power of the court, which inevitably makes people feel that the victim is passive.

### 3. How the Structure of the ICC Reparations System Conflicts with Victim Reparations in Practical Terms

#### 3.1. Small Compensation

Even though the criminal courts provide victims with many ways to obtain compensation, the actual compensation is too small to satisfy victims. The previous section mainly analyzes the ICC's care for victims, but the tension between victims and the ICC's compensation is also an important aspect. Under the current ICC system, there are two ways for victims to participate in the court. One is to participate in the courtroom to express one's views and concerns during the proceedings. The second is to participate in the compensation trial stage. Nonetheless, both approaches are extremely limited. In addition to political and international factors, the victim may not be qualified as a victim, and the right to compensation with the victim's qualification is not well protected. As mentioned in the previous section, most victims qualify as victims, although the perpetrators are ultimately convicted of most of their primary crimes. However, prosecutors' focus on high-ranking officials, heads of state, leaders, and those most responsible for crimes will result in a large number of victims' compensation being paid by only a few perpetrators. Furthermore, the perpetrators are mostly poor and are therefore unlikely to be able to afford the huge compensation. Even if the ICC was able to allow the TFV to replace the offender's compensation, the efficiency of the TFV's execution of the compensation order is very slow. Secondly, the TFV's compensation direction is usually from the perspective of the court rather than meeting the needs of the victim. It is regrettable to see victims, who, having spent a long time in court asking for compensation, enduring the trickier aspects of the court process, only receive USD 250 in damages individually. Most of the victims

expect the international community, that is, the International Criminal Court, to provide them with adequate compensation or the ability to face the hardships of life resulting from the crimes that they have suffered.

### 3.2. Silence of Courts on Victims' Rights of Compensation

The ICC has hardly given enough financial compensation, nor has it received adequate attention and respect in the proceedings. In particular, the eligibility of victims, how victims are protected, and the type of compensation that victims receive are determined by the courts. Another example is that the victim does not decide what happens in a lawsuit; they are never the party in control. On the other hand, even in the documents of the International Criminal Court, it is common to see judges taking care of the rights of victims, but only as an act of their own will. Furthermore, in court documents, most victims' rights are realized by providing victims' applications, and courts rarely use the term "victims' rights". This means that the courts are silent on the rights of victims, and the courts are always worried about victims, affecting the length of the trial or the workload of the court. In this context, the court considers that victims are a burden on the court. Courts rarely give victims a choice of how they wish to be compensated. For example, in the Lubanga case, the Trial Chamber ignored individual applications and instead offered compensation by way of collective compensation (International Criminal Court 2015b), which disappointed some victims. However, the Chamber stressed that even international human rights did not recognize the human right to consider individual compensation claims. Perhaps from this statement, the court does seem to treat the victim as a "problem" to be solved rather than as an individual (Zegveld 2019, (n48)). Hence, although the court provides victims with many ways of achieving rights protection, in court documents or the Rome statute, the term "victims' rights" is rarely mentioned, or the rights of the victim, with such language being rare, all depending on whether the judge will unilaterally decide the victim's related treatment. This undoubtedly shows the silence of the court on the rights of victims.

### 3.3. The ICC's Model of Judicial Administration Is Unfavorable to Victim Compensation

Another source of concern is that the ICC's model of judicial administration remains optional (Torrens 2020). First, the ICC operates with a regional focus rather than having universal jurisdiction in a global region. In particular, in the roughly 15 years that the court has operated, it has never punished a non-African person. In other parts of the world where conflicts and international crimes are more evident, the court has remained silent; for example, British military personnel's alleged war crimes in Iraq and Afghanistan (Braithwaite and Wardak 2013; Wardak and Braithwaite 2013), the 2006 Israeli–Lebanese conflict (Sharp et al. 2006), and the continuing Israeli–Palestinian conflict. Despite the seriousness of these incidents, the courts have no intention of prosecuting them. Although some non-African situations (Venezuela, Palestine, and Iraq) are currently under preliminary investigation by the prosecutor, it does not mean that the prosecutor will officially investigate since it is difficult to distinguish the intentions of the prosecutor (Jim 2014). Moreover, according to the Rome Statute (UN General Assembly 1998), the prosecutor must get the authorization of the Pre-trial Chamber first, then the prosecutor may initiate the investigations unless the prosecutor has initiated the investigation at the request of the Security Council in which case the intervention of the Pre-trial Chamber is not required (Garkawe 2001). Thus, the Prosecutor's right to prosecute is also subject to certain limitations and controls (Caesius 1999, p. 21).

The contradiction between the two also lies in the provisions on victims stipulated in the Rome Statute: the discourse is close to declarative expression, but there are few provisions on the actual and specific content. For example, Article 68 (3) of the Rome Statute (UN General Assembly 1998) uses vague language for how victims can participate in trials. It states only that the court should allow victims to raise their concerns in a manner that does not prejudice the rights and fairness of the accused at the stage of the proceedings that the court deems appropriate. The Chamber, therefore, found that the article allowed

the victim to participate at any time without undue harm to the accused. The court held that they had an arbitrary right to refuse to participate as they saw fit. As a result of the vagueness of the Rome Statute regarding victims, the rights of victims are usually dealt with on a case-by-case basis by the courts (Kersten 2016). In the Chamber's question on whether to authorize 20 victims to participate in the tribunal, the prosecutor sought to exclude all victims who had no or insufficient evidence or whose evidence was flawed. However, the court stated that only a preliminary examination of the victim's application would be conducted. From this case, it can be seen that the prosecutor does not want the victim to participate in the court, so it is extremely strict on the qualification of the victim. Thus, it can be seen that vague provisions on rights will become a means for other parties to block the realization of victims' rights. In addition, the victim has successfully become a party to the tribunal to participate in the tribunal, but the court still needs to reduce the degree of arbitrary justice. Because victims play an important role in the compensation phase of a trial, they will form a three-way balance of power with the defendant and the lawyer. The Rome Statute provides for victims to be able to testify at sentencing hearings, which can reduce a defendant's sentence if he or she apologizes to the victim and accepts responsibility for his or her actions. The provision facilitated reconciliation between the victim and the accused. This provision is also a kind of restorative justice. At present, the court has only four existing rulings, so it is difficult for victims to find relevant rulings from precedents. Even the pre-trial and trial Chambers often have inconsistent verdicts. For example, the image of the child soldier, in particular, has received conflicting portrayals in the cases of Lubanga and Ongwen[35]. In Lubanga, the Trial Chamber found that Lubanga children were weak, passive, and permanently affected by victimization from which they could not recover (Kersten 2016); in Ongwen, the pre-trial tribunal denied that the child soldier was permanently impaired and held that the victim's status would be lost in adulthood. Obviously, the court held completely different positions on the same issue in the two cases, which shows the arbitrariness of the court's judgment. Since the ICC has not tried many cases so far and the relevant laws and regulations are sporadic, the judgment results are highly arbitrary and uncertain. However, courts should provide predictability to victims so that arbitrary decisions do not interfere with justice. Hence, victims urgently need the ICC to formulate a set of principles for victim compensation. In the literature on the ICC and victims, scholars pay more attention to the system and ignore the importance of victim psychotherapy. Of course, the psychological treatment of victims needs to be scientific and rigorous, and the court cannot arbitrarily speculate on whether a certain behavior is meaningful for psychological treatment (McCarthy 2012).

*3.4. Conclusions*

Even though the criminal courts provide victims with many ways to obtain compensation, the actual compensation is too small to satisfy victims. The previous section mainly analyzes the ICC's care for victims, but the tension between victims and the ICC's compensation is also an important aspect. Under the current ICC system, there are two ways for victims to participate in the court. One is to participate in the courtroom to express one's views and concerns during the proceedings. The second is to participate in the compensation trial stage. Both approaches are extremely limited. Although most victims qualify as victims, a large number of victims' compensation being paid by only a few perpetrators. Furthermore, the perpetrators are mostly poor and are therefore unlikely to be able to afford the huge compensation. Even if the ICC was able to allow the TFV to replace the offender's compensation, the efficiency of the TFV's execution of the compensation order is very slow. Secondly, the TFV's compensation direction is usually from the perspective of the court rather than meeting the needs of the victim. Further, the ICC has hardly given enough financial compensation, nor has it received adequate attention and respect in the

---

[35] The case of the Prosecutor v. Dominic Ongwen, Trial Chamber IX, 6 May 2021, ICC-02/04-01/15 (International Criminal Court 2005).

proceedings. The victim does not decide what happens in a lawsuit; they are never the party in control. On the other hand, even in the documents of ICC, it is common to see judges taking care of the rights of victims, but only as an act of their own will. Furthermore, in court documents, most victims' rights are realized by providing victims' applications, and courts rarely use the term "victims' rights". The courts are silent on the rights of victims; the court will often worry about victims affecting the length of the trial or the workload of the court. Courts rarely give victims a choice of how they wish to be compensated. Perhaps from this statement, the court does seem to treat the victim as a "problem" to be dealt with rather than as an individual. Although the court provides victims with many ways of achieving rights protection, in court documents or the Rome statute, the term "victims' rights" is rarely mentioned, or rights of the victim, with such language being rare, all depending on whether the judge will unilaterally decide the victim's related treatment. In the meantime, the ICC's model of judicial administration remains optional, which is unfavorable to victim compensation. Although the prosecution is conducting a preliminary investigation into non-African regions, it is not clear whether it will decide on a formal investigation and indictment. In addition, in the Rome Statute's provisions on victims, the discourse is closer to declarative expressions and less prescriptive in terms of practical details. This kind of phenomenon has a more or less negative impact on the protection of victims' rights. Because of the vagueness of the Rome Statute's provisions on victims, the rights of victims are usually dealt with by the court on a case-by-case basis. Vague rights provisions may become a means for others to impede the fulfillment of victims' rights. Moreover, as the ICC has heard fewer cases so far and the relevant laws and regulations are fragmented, the outcome of judgments, to some extent, is arbitrary and uncertain. Victims, therefore, urgently need the ICC to develop a set of principles for victim reparations.

## 4. Potential Obstacles and Gap between Criminal Trial Reparations and Victims

### *4.1. Victims' "Social Death" Poses a Challenge to ICC's Progress on Reparations*

The "social death" of victims makes the ICC treat them as a "problem". First, the reason that the ICC sees victims as a problem is because it is still an institution focused on retaliatory justice. Although the ICC has evolved such practices as reparations and victim participation in court, it is essentially focused on retributive justice. The ultimate purpose of the court is to punish the crime, not to compensate the victim, so the victim inevitably becomes a burden to the court. Second, the perpetrator is extremely cruel, and such cruelty to victims would lead to a general feeling for humanity; for example, the victim is often described as someone on the outer edges of society, compared to cockroaches, rats, or monsters, which seems to suggest a victim being turned into an object of revulsion, with an attendant loss of humanity (ReliefWeb 2010). James Waller (2007) once called this phenomenon "the social death of victims". Furthermore, the victim may also be seen as "bad". As an illustration, in transitional justice, scholars have found a role for innocence in the construction of victim identities; that is, Geis (1990) has argued that the public and political accept the idea that victims are good people; they are victimized by bad people. In practice, however, the degree of "innocence" forms a hierarchy of victimhood (McEvoy and McConnachie 2013, p. 501), and there is even a distinction between "good" victims and "bad" or 'impure' (Madlingozi 2007) victims (Madlingozi 2007, p. 111). This kind of hierarchy of victims corresponds to the formation of subjective perceptions of the "justifiability" of the suffering and pain suffered by some of the victims (McEvoy and McConnachie 2012, pp. 528–30), who seem to have an external subjective perspective to 'justify' the suffering of victims. Likewise, the victim's injuries are perceived as self-inflicted in some way. Of course, it is undeniable that in the case of homicide, for example, Wolfgang argues that, indeed, in some homicide offenses, victims and killers are sometimes 'mutual participants in homicide' and that some victims may precipitate their murder (Wolfgang 1957). Victims sometimes engage in carelessness, impudence, provocation, misconceptions, and other behaviors, which sometimes become factors that lead to victim blaming (O'Connell 2008, p. 94). With this kind of guilt, the victim will gradually turn on himself or herself and see himself or herself as the "source" of

all evil. Victims are defined as being outside the perpetrator's universe of moral obligation. In this context, the court also treats the victim as some sort of "problem" arising from the crime. Victims are seen by society as "products" outside the scope of their moral obligations. To date, the ICC has always involved the de-individualization of its victims, that the victims are kept together for a long time, in their non-human or consistent status. Thus, the social death of victims has led to the ICC still treating victims as a problem. However, many victims lack rights resources, some have also lost relevant life skills or abilities, etc., and they do need the support and assistance of the ICC (McEvoy and McConnachie 2013, p. 499). The ICC should avoid the long-term perpetuation of this perception, and the court should become a force for reducing the process of death in the victim's society and reducing society's denial of the individual victim (Ezennia 2016).

### 4.2. Cultural Background Factors and Restrictions on "Expression" Clog up the Possibility of Compensation

As noted above, much effort has been made for victims to participate in the courts. However, the courts impose expressive limits on victims. In the Rutaganda case, the court made it clear that it would consider the social and cultural factors of witnesses when considering their testimony.[36] In the Akayesu case[37], the court found that in Rwandan culture, people do not always answer questions directly. The silence does not imply approval, and the answers need to be deciphered by the courts in context. During the trial, the court repeatedly asked whether the term "Inyenzi" stood for "cockroach", but witnesses were reluctant to confirm it. The court ultimately decided to take cultural factors into account and not to infer against the witness's silence. In this case, the communication between the court and the witness was indeed difficult, especially when the witness did not cooperate by answering the court's questions due to cultural background factors. The contradiction between the two produces an extreme result: either the court accepts the influence of cultural background, or it refuses to consider it (Sander 2019). In the Civil Defence Forces (CDF) trial at the Special Court for Sierra Leone (SCSL), prosecutors argued that the defendants used religion to manipulate local cultural beliefs and as a weapon to kill civilians. It is true in a certain sense, but even though the court had considered it, there was no substantial proof that the defendants had mystical powers over the victims. It was concluded that the defendant did not substantially control the victim. The ruling clearly shows that the court chose to remain silent on the evidence and make its judgment based on Western thinking. It is sad and grossly unfair to the victims.

Due to the factors of cultural differences, the court cannot use the cultural background of the victims for their thinking. For example, religious factors can greatly influence and control victims in deprived areas, but the court finally concluded that there was "no actual control" based on the cultural background of the court. Even if the court takes due account of the victim's cultural background and expression, the Western culture of the court system itself may, to some extent, also play a small role in influencing the details of the trial. Moreover, in addition to cultural differences that courts themselves are aware of, there are also unrecognized differences in communication between courts and victims. The ability to accurately recognize the communication problems caused by cultural differences involves the sensitivity of the court to the local cultural context in which witness testimony is provided (Sander 2019). It raises another issue that requires attention: the authenticity of the witness's testimony. Cultural factors may cause a witness to logically fail to understand a leading question from a prosecutor or a defense lawyer and give an incorrect answer or cause the court to misunderstand an answer. Since international criminal justice agencies lack the funds for fact-checking on the ground and the conditions to visit certain areas, it is easy for witnesses to alter the truth of their testimony or lie for

---

[36] The Prosecutor v. Georges Anderson Nderubumwe Rutaganda, Trial Chamber I, 6 December 1999, ICTR-96-3-T.
[37] The Prosecutor v. Jean-Paul Akayesu, Appeals Chamber, 1 June 2001, ICTR-96-4-A.

certain benefits. Differences in cultural factors cause problems in communication between courts and victims (Sander 2019, p. 6).

### 4.3. Does the ICC Hope to Improve Its Attitude toward Victims's Compensation?

The ICC as a whole prosecutes for the safety of the international community, not for victims. Although the ICC has attracted much attention for its care and compensation for victims, few victims have actually received compensation. None of the four compensation judgments made by the main ICC so far has been enforced, meaning that victims have not received any compensation from the perpetrators. Combined with the collective compensation and psychological counseling measures commonly used by the ICC, it can be concluded that the ICC actually focuses most of its energy on helping victims and victim communities restore their normal lives. Secondly, the ICC still regards victims as a tool to punish international crimes in terms of the difficulty in initiating prosecution proceedings for them and their right to participate in the process from prosecution to trial being unilaterally decided by the court. The ICC aims to pursue criminal responsibility for the security and stability of the international community rather than initiating prosecution proceedings for individual victims or groups.

This aim directly leads to the difficulty for the victim to become a subject with independent litigation rights and status, which is exactly the reason why the victim wishes to prosecute the crime at the ICC. Thus, fundamentally, the ICC runs counter to the purpose of victims, and the ICC's attitude toward the victims can only be limited to compensation that is given priority. Collective personal compensation is complementary to protect the victim's rights and ensure they continue to improve, but not for the victims to change course, being unable to ensure that the victim has the lawsuit main body status and main body status of the corresponding rights and respect.

### 4.4. Conclusions

Social death allows the victim to be defined outside the scope of the perpetrator's moral obligations. Furthermore, the victim may also be seen as "bad". In practice, the degree of "innocence" forms a hierarchy of victimhood, and there is even a distinction between "good" victims and "bad" or 'impure' victims. The "social death" of victims causes the ICC to treat them as a "problem". In this case, the court also treats the victim as some kind of "problem" arising from the crime, and the victim remains de-individualized in the court for a long time, in a non-human or consistent position. The ICC should avoid perpetuating this notion, and the court should be a force for reducing the process of death in the victim's society and the rejection of the individual victim. Otherwise, in the long term, the victim, too, may become disillusioned with the ICC. The gap between ICC and victims also exists in the courts that impose expressive limits on victims. Because of cultural differences, the court cannot think in terms of the victim's cultural background, which means that the court itself does not seem to punish crimes from the victim's point of view. Instead, crime is punished from the perspective of the "justice" of the court itself, which represents a deviation between the justice of the court and that of the victim. This raises a fundamental question of whether the ICC is prosecuting for the safety of the international community rather than the victims. In combination with the ICC's usual measures of collective reparations and psychological counseling, it can be seen that the ICC focuses most of its energy on helping victims and victim communities to rehabilitate. Second, the ICC still sees victims as tools to punish international crimes because victims have difficulty initiating prosecution proceedings and have the right to participate in a process from prosecution to trial unilaterally decided by the court. The purpose of the ICC is to pursue criminal responsibility for the security and stability of the international community, not to prosecute individuals or groups of victims. Therefore, the ICC fundamentally defeats the purpose of the victims. Thus, there are several potential obstacles between victim compensation and ICC, thus forming a huge gap between the two.

## 5. Conclusions

Both types of justice have deeply influenced international criminal justice institutions and early international criminal justice, and the weight of both types of justice has changed over time. Early international criminal justice as a whole focused heavily on retributive justice, which also influenced early ICC, although early military tribunals, such as the ICTY and ICTR, made limited progress on victim reparations. Restorative justice gradually influenced the early ICC, which gradually shifted from retributive justice to various restorative measures for victims and became a major highlight and feature of the development of the ICC. Gradually, the courts also began to become thoughtful and considerate in the matter of victim compensation. Victims can not only participate in compensation trials, become a party to the court, and receive individual compensation, but they can also receive something akin to collective compensation to help them and their communities return to normal life, but courts rarely provide individual compensation, the court gave symbolic compensation, which is ostensibly personal compensation, but it is still similar to collective compensation in essence. Over time, the ICC has developed a second type of justice, participatory justice, and expanded the scope of victim eligibility, which means that more victims will receive compensation in the future. The ICC allows victims to become a party in reparation trials and to be witnesses, which is very meaningful for compensation because victims' participation and compensation are related and complementary to each other, as participation of the victims will bring a lot of help to the issue of compensation. Unfortunately, however, the implementation of reparation orders is somewhat difficult due to the fact that some states do not accept the ICC or agree that the implementation of reparation orders is also subject to difficult and slightly more complicated steps. Most states do not outright refuse to cooperate but rather fail to act or procrastinate. In practice, the ICC's time and effort in arresting, seizing, freezing funds, and communicating with the State is difficult and without guarantee of success. Additionally, even if the offender's property could be used to enforce a reparation order, the amount that would be realized for the individual victim would be extremely limited. However, since the Court, as an independent international organization with the power to react to non-cooperative states, is also actively contributing to the evolution of the issue of enforcement of reparation orders, the expansion of the scope of victim compensation by the Court may also be a trend in the future development of reparation orders.

For two approaches to compensation, collective and individual compensation methods, the use of collective reparations avoids the cumbersome procedures and burdensome tasks of the International Criminal Court, while individuals who are not eligible for reparations to victims are more likely to provide collective reparations. Collective reparations are a form of reparation, although they are not referred to by the Court as individual reparations. This is essentially the application of the court's flexibility to collective reparations. Flexibility is also reflected in the fact that different victims have different needs and are compensated differently by the courts. Fortunately, collective reparations can help restore the neighborhoods where victims live. This may mean restoring infrastructure to the neighborhoods where the victims live and providing educational services for the victims' children. This approach can greatly increase the satisfaction of victimized communities with the ICC. The disadvantage is that the court's facilitation of collective reparations, as well as its consideration of the cost of reparations, may vary from region to region.

Courts and local jurisdictions sometimes provide less or no reparations to distant victims to facilitate implementation. Even so, however, as a large proportion of victims require financial assistance, victim groups have shown a strong preference for financial compensation, which is largely difficult to meet using collective reparations. Individual reparations, on the other hand, rarely occur, and when they do, they are too small to cover the needs of the victims. Further, since the influence of victims on the prosecutor is limited and essentially confined to giving evidence and filing complaints, the court provides other remedies for victims. The court of the qualification of the victim is not strict, only a superficial review, which is very favorable to victims' compensation matters. For instance,

the criteria for determining the victim should be a preliminary assessment, without an in-depth assessment, and it is not strict in its determination of victim eligibility, which is in a lenient attitude. As long as the credible statement of the victim and the presumption of fact are reliable, the fact of the victim's guilt is not a matter for the victim. As long as the credible statement of the victim and the presumption of fact are reliable, the fact of the injury can be presumed. The Court takes into account the fact that it is often difficult for the victim to present complete and sufficient evidence to prove consistency. The ICC has adopted a flexible approach to review, while VPRS, OPCV, PIOS, and other court organizations are available to assist victims, provide insight into the situation and needs of victims, and advocate for the ICC among local people so that victims can learn about and turn to the ICC for help, details that demonstrate the victim-friendliness of the ICC. However, VRPS has been relying partly on intermediaries to fulfill some of its duties, and it is difficult for some victims to reach the slightly distant VPRS and to get reasonable help from intermediaries, and it is difficult to say whether this indirect communication is problematic or not. Since VPRS evidence may be gathered using outreach agencies, language experts do not have the opportunity to communicate directly with victims, so miscommunication remains an issue. The VPRS also simply does not have enough money to conduct extensive checks on the evidence it collects. Furthermore, due to victims usually appearing in groups, the court selectively involves a very small number of victims as individuals. Scholars have argued that the victims are the ICC's "Brand". The ICC hopes to shape an institutionalized concept of the ideal victim for all member states, making this institutionalized concept a "spokesperson" for global justice and a brand for the ICC to promote and enhance its influence. The ICC should be the core of the international community's condemnation of international crimes. It should not be developed to favor the court's preferences. This will damage the stability of the international community and the responsibilities entrusted to the ICC by that community. The question of the court system may not represent the slanting of the court's objectives, but at least measures should be developed as far as possible to avoid such deficiencies. In addition to the detailed compensation, compensation funds are also very extensive. The establishment of the TFV indicates that the court actively supports the victim's compensation and has made some efforts. Collective compensation will involve the cooperation of multiple governments, which will take a lot of time, and it is uncertain whether negotiations between governments and courts will be successful. Individuals are unable to initiate collective compensation on their initiative, nor are they able to control the process and procedures. The victim's only option is to defer to the courts, even if that means waiting indefinitely, whereas TFV can greatly increase the scope for victims to receive compensation. However, there is an irreconcilable contradiction between the ICC and the TFV in the execution of compensation orders. To date, the ICC's compensation order has not been implemented (Ingadottir 2003, p. 14). In fact, the statute of the ICC refers only to the ICC in terms of financing, so the TFV and the court's budget system are separate and mutually exclusive. Even if the courts share some of the same goals as the TFV, it still depends on what the courts ultimately decide.

The structure of the ICC reparations system conflicts with victim reparations on several fronts. Firstly, although ICC's compensation is thoughtful and provides many avenues for obtaining compensation, the actual amount of compensation is very small. The tension between victims and the ICC's compensation is also an important aspect. Although victim participation in trials can be very beneficial to victims in obtaining compensation, the ways in which victims can participate in trials are very limited in both cases and because prosecutors usually focus on the few most responsible, large numbers of victims are often compensated by only a few perpetrators. Although victim participation in trials can be very beneficial to victims, both modalities of victim participation are very limited, and because prosecutors usually focus on the few who bear the greatest responsibility, the result is that compensation for a large number of victims is usually paid by only a few perpetrators. Even if TFVs are able to make reparations in lieu of perpetrators, the efficiency of TFVs in enforcing reparation orders is very slow, and reparations are usually made from the

perspective of the court rather than meeting the needs of the victims. The ICC has hardly given enough financial compensation, nor has it received adequate attention and respect in the proceedings. Victim eligibility, how victims are protected, types of reparations, how help is provided, etc., are all determined by the courts. The victim does not decide what happens in a lawsuit; they are never the party in control.

Judges take care of the rights of victims, but only as an act of their own will; the courts are silent on the rights of victims. Although the court provides victims with many ways of achieving rights protection, in court documents or the Rome statute, the term "victims' rights" is rarely mentioned, or rights of the victim, with such language being rare, all depending on whether the judge will unilaterally decide the victim's related treatment. This undoubtedly shows the silence of the court on the rights of victims. Further, the ICC's model of judicial administration is unfavorable to victim compensation. ICC may have no intention of prosecuting even in areas where international crimes are widespread. Although some non-African situations (Venezuela, Palestine, and Iraq) are currently under preliminary investigation by the prosecutor, it does not mean that the prosecutor will officially investigate since it is difficult to distinguish the intentions of the prosecutor. The powers of the Prosecutor are also limited by the court, and in most cases, the prosecutor must be authorized by the Pre-Trial Chamber to commence an investigation. The contradiction between the two also lies in the provisions on victims stipulated in the Rome Statute. Since the ICC was established under the Rome Statute, the activities of the ICC are almost exclusively based on the Rome Statute, which uses rather vague language as far as the rights of victims are concerned. The rights of victims are usually dealt with by the Court on a case-by-case basis, and as a result, judgments of Trial Chambers are often inconsistent in their rulings, which shows a certain degree of arbitrariness and uncertainty in judgments on this issue.

In addition to the problems with the ICC's reparations mechanism, a number of hidden potential obstacles and gaps between criminal trial reparations and victims still exist and have a considerable impact on the issue of victim reparations. First and foremost, the ICC has an underlying tendency to view victims as a burden, a problem. The "social death" of victims makes the ICC treat them as a "problem". To date, the ICC has always involved the de-individualization of its victims, that the victims are kept together for a long time, in their non-human or consistent status. However, many victims lack rights and resources, some have also lost relevant life skills or abilities, etc., and they do need the support and assistance of the ICC. The ICC should avoid the long-term perpetuation of this perception, and the court should become a force for reducing the process of death in the victim's society and reducing society's denial of the individual victim. Furthermore, cultural background factors and Restrictions on" expression" clog up the possibility of compensation. As noted above, much effort has been made for victims to participate in the courts. However, the courts impose expressive limits on victims. Due to the factors of cultural differences, the court cannot use the cultural background of the victims for their thinking. Moreover, in addition to cultural differences that courts themselves are aware of, there are also unrecognized differences in communication between courts and victims. The ability to accurately recognize the communication problems caused by cultural differences involves the sensitivity of the court to the local cultural context in which witness testimony is provided. Differences in cultural factors cause problems in communication between courts and victims, whereas because the ICC as a whole prosecutes for the safety of the international community, not for victims. Although the ICC has attracted much attention for its care and compensation for victims, few victims have actually received compensation. None of the four compensation judgments made by the main ICC so far has been enforced, meaning that victims have not received any compensation from the perpetrators. The ICC actually focuses most of its energy on helping victims and victim communities to restore their normal lives. Secondly, the ICC still regards victims as a tool to punish international crimes in terms of the difficulty in initiating prosecution proceedings for them and their right to participate in the process from prosecution to trial being unilaterally decided by

the court. The ICC aims to pursue criminal responsibility for the security and stability of the international community, rather than initiating prosecution proceedings for individual victims or groups. This aim directly leads to the difficulty for the victim to become a subject with independent litigation rights and status, which is exactly the reason why the victim wishes to prosecute the crime at the ICC. Fundamentally, the ICC runs counter to the purpose of victims, and the ICC's attitude toward the victims can only be limited to compensation that is given priority to.

**Funding:** This research received no external funding.

**Institutional Review Board Statement:** Not applicable.

**Informed Consent Statement:** Not applicable.

**Data Availability Statement:** Data sharing not applicable. No new data were created or analyzed in this study. Data sharing is not applicable to this article.

**Conflicts of Interest:** The author declares no conflict of interest.

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
