# Peer review of "The Gap between the International Criminal Court and Victims: Criminal Trial Reparations as a Case Study"

_laws_

Round 1
Reviewer 1 Report
This article would be much clearer if the author(s) referred to International Criminal Justice more broadly in their introduction. Stating that the ICC began with Nuremberg is confusing as the ICC was established through the Rome Statute, and was not a multi-national military tribunal established at the end of a great power war. Also, analytical categories like restorative justice v. retributive justice should be consistently applied.
This could be done by more explicitly defining and setting out these two approaches to justice early on, then proceeding with the analysis through a discussion of reparations in international justice.
Also, more engagement with scholarship on victims in international justice/transitional justice would help provide more conceptual clarity around this discussion (i.e. McEvoy).
This submission should be proofread carefully - there are some section headings that contain grammatical errors. For example, "How is the idea of compensation change"
Reviewer 2 Report
focus on one topic . this work deals with many different issues and it is not clear how they relate to each other
improvement is needed
Reviewer 3 Report
This draft is overall well structured, detailed and more or less comprehensively researched. The conclusion is rather convincing and logical; the purpose of the paper is rather clear. The topic itself is also relevant and to the point: the issue of how well rights of victims have so far been provided at the ICC remains topical. However, the text as presented has multiple issues and problems to be fixed. First, there are many grammar / language issues and gaps + sentence building problems. Some (though not too many) sentences are simply incomprehensible for the reviewer (e.g., I do not understand what the first sentence in lines 170-171 means). There are also mistakes with declarative vs. interrogative sentences, for example, improper overuse of the verbs such as "do", "does", etc. as well as lack of exclamation marks where needed including in the section/subsection titles. Second, many citations especially to the ICC case law are missing; this needs to be mended. Furthermore, there is some speculation in certain subsections that I'd recommend to avoid. E.g., subsection 4.2. appears to be speculative and perhaps even biased - statements to the effect that there is evidence (no reference to the figures, btw) that ICC is only promoting and protecting Western interests seems strongly judgmental. Such a statement needs to be either much more fully supported with concrete detail / data, or to be dropped whatsoever. This seems especially surprising given such a well-written and supportive overall conclusion; I'd recommend that the author avoids contradictory or even biased statements like that. It spoils the overall impression from an otherwise well-structured and argued academic piece. So, I suggest the paper is published but only after concrete revisions based on the feedback above and upon a careful English proof-reading.
Already provided above in the first field.
Round 2
Reviewer 1 Report
This paper is much improved and I am now happy to recommend publication
This paper's English is also much improved and I am now happy to recommend publication
Author Response
Dear reviewer,
I sincerely thank you for your precious suggestions and professional work on the article. I am very grateful for your suggestion and I wish you the very best!
Warm regards
Reviewer 2 Report
I would try to better state the research questions at the beginning and to link the conclusions reached to the arguments presented in the paper
good
Author Response
Dear reviewer,
I sincerely thank you for your precious suggestions and professional work on the article. I am very grateful for your suggestion and have revised each part of the article very carefully based on that suggestion. I followed all your suggestions and based on the suggestions, I have better stated the research question of the article, the research questions, and the research content of each chapter in the beginning, and introduced the research objectives of each chapter so that the connection of each chapter is clearer. Then, I slightly changed the titles of some of the sub-chapters so that the chapters would be better connected and the titles would be clearer. Based on your suggestions, I have also clarified the topic sentence of each paragraph in order to make the conclusion and each paragraph more connected. I revised the conclusion of each chapter to link the conclusions reached to the arguments presented in the paper.
Thank you and I wish you are the very best!
Best regards,